# Exploiting DNA methylation in cassava under water deficit for crop improvement

**Jorge Luís Bandeira da Silva Filho**[1☯], **Rosa Karla Nogueira Pestana**[2☯], **Wilson José da Silva Júnior**[1☯], **Maurício Antônio Coelho Filho**[2], **Claudia Fortes Ferreira**[2], **Eder Jorge de Oliveira**[2]*, **Ederson Akio Kido**[1]*

1 Laboratório de Genética Molecular de Plantas, Departamento de Genética, Universidade Federal de Pernambuco, Recife, Brazil, 2 Embrapa Mandioca e Fruticultura, Rua da Embrapa, Cruz das Almas, BA, Brazil

☯ These authors contributed equally to this work.
* ederson.kido@ufpe.br (EAK); eder.oliveira@embrapa.br (EJO)

## Abstract

DNA methylation plays a key role in the development and plant responses to biotic and abiotic stresses. This work aimed to evaluate the DNA methylation in contrasting cassava genotypes for water deficit tolerance. The varieties BRS Formosa (bitter) and BRS Dourada (sweet) were grown under greenhouse conditions for 50 days, and afterwards, irrigation was suspended. The stressed (water deficit) and non-stressed plants (negative control) consisted the treatments with five plants per variety. The DNA samples of each variety and treatment provided 12 MethylRAD-Seq libraries (two cassava varieties, two treatments, and three replicates). The sequenced data revealed methylated sites covering 18 to 21% of the *Manihot esculenta* Crantz genome, depending on the variety and the treatment. The CCGG methylated sites mapped mostly in intergenic regions, exons, and introns, while the CCNGG sites mapped mostly intergenic, upstream, introns, and exons regions. In both cases, methylated sites in UTRs were less detected. The differentially methylated sites analysis indicated distinct methylation profiles since only 12% of the sites (CCGG and CCNGG) were methylated in both varieties. Enriched gene ontology terms highlighted the immediate response of the bitter variety to stress, while the sweet variety appears to suffer more potential stress-damages. The predicted protein-protein interaction networks reinforced such profiles. Additionally, the genomes of the BRS varieties uncovered SNPs/INDELs events covering genes stood out by the interactomes. Our data can be useful in deciphering the roles of DNA methylation in cassava drought-tolerance responses and adaptation to abiotic stresses.

## Introduction

Cassava (*Manihot esculenta* Crantz) is an economically relevant crop providing food security to more than 800 million people worldwide [1]. This crop is widely cultivated in tropical and subtropical climates [2] and is well adapted to the semi-arid climate of the Northeast region of Brazil, growing under conditions of direct sunlight and water restriction, representing an alternative income for small farmers ensuring their livelihood [3].

**Data Availability Statement:** All datasets generated for this study can be found in the article,

supplementary material, and Figshare (https://doi.org/10.6084/m9.figshare.21330708.v2).

**Funding:** Funding • Jorge Luís Bandeira da Silva Filho: CAPES (Coordenação de Aperfeiçoamento de Pessoal de Nível Superior). Grant number: 88887.814148/2023-00 • Rosa Karla Nogueira Pestana: FAPESB (Fundação de Amparo à Pesquisa do Estado da Bahia). Grant number: 1357/2015 • Claudia Fortes Ferreira: CNPq (Conselho Nacional de Desenvolvimento Científico e Tecnológico). Grant number: 302145/2022-2 • Eder Jorge de Oliveira: CNPq (Conselho Nacional de Desenvolvimento Científico e Tecnológico). Grant number: 409229/2018-0, 442050/2019-4 and 303912/2018-9; FAPESB (Fundação de Amparo à Pesquisa do Estado da Bahia). Grant number: Pronem 15/2014 • Ederson Akio Kido: CNPq (Conselho Nacional de Desenvolvimento Científico e Tecnológico). Grant number: 311894/2017-8. • This work was partially funded by UK's Foreign, Commonwealth & Development Office (FCDO) and the Bill & Melinda Gates Foundation. Grant INV-007637. Under the grant conditions of the Foundation, a Creative Commons Attribution 4.0 Generic License has already been assigned to the Author Accepted Manuscript version that might arise from this submission. • The funder provided support in the form of fellowship and funds for the research, but did not have any additional role in the study design, data collection and analysis, decision to publish, nor preparation of the manuscript.

**Competing interests:** The authors have declared that no competing interests exist.

Embrapa Mandioca e Fruticultura detains one of the largest cassava germplasm collection and genetic breeding program in the world. Specifically, concerning cassava, the release of new varieties is a constant demand to mitigate problems faced by farmers. The BRS Formosa variety (bitter cassava), resistant to *Xanthomonas phaseoli* pv *manihotis*, is a well-adapted variety to cultivation in the semiarid region (high temperature and low rainfall ~280–800 mm); a drought prone region covering the Northeast of Brazil [4]. In contrast, the BRS Dourada, a sweet cassava variety released by Embrapa´s breeding program, has high culinary quality of the roots, as well as the high root yield, low cooking time, and high beta-carotene content [5], but does not thrive in semi-arid regions.

With the intensification of climate change, drought prone areas should increase, compromising crop yields and agricultural lands [6]. As water supply and arable land become scarcer, the development of drought-tolerant plants and the incorporation of marginal lands becomes imperative. Plants under water stress often reduce root proliferation, leaf size, and stem length, disturbing plant-water relationships and reducing water use efficiency [2]. However, plants have a variety of processes to minimize drought stress, whereas epigenetic mechanisms, such as DNA methylation, modulate gene expression profiles aiming at the immediate stress response [7–9].

DNA methylation of the cytosine of the CG dinucleotide and CNG and CNN sequences, practically occurs in all organisms, including plants and animals [10]. This hereditary epigenetic mark is a critical factor in the regulation of chromatin structure and gene expression, mainly altering the stability and positioning of nucleosomes; the fundamental unit of chromatin, affecting the accessibility of DNA to regulatory proteins or protein complexes involved in DNA replication, repair, and transcription [7]. Thus, DNA methylation influences several biological processes, such as plant growth and development, flowering, gene expression, and environmental stress responses [8].

During methylation process in plants, the methyltransferase enzyme incorporates methyl groups to the cytosine. Basically, the DNA methyltransferase type I (MET1) performs GC methylation in newly synthesized DNAs, also necessary to maintain methylation patterns in repetitive and single copy sequences during gametogenesis [11]. In plants under normal conditions, the proportion of methylated cytosines to the total values varies from 20% to 30% [12].

Some methylation-related changes in the genome of plants responding to stress are temporary, while others may be stable and heritable for future generations, allowing descendant plants to manage those challenges in the future [13]. The DNA methylation profile is at some level engaged in the genetic responses of plants to abiotic stresses [14]. During stress responses, changes in DNA methylation profiles become linked to changes in the regulation of genes involved in the stress-response process and interactions of regulatory pathways [15]. In drought-tolerant rice plants, genes associated with differentially methylated regions were also associated with gene ontology terms related to stress response, programmed cell death, nutrient reservoir activity and constitutive water stress tolerance [16]. Furthermore, apparently tolerant plants have a more stable methylation profile than sensitive plants, presenting fewer drought-induced differentially methylated regions [16]. In the present work, we compared the methylation profile of two cassava varieties [BRS Formosa (bitter) and BRS Dourada (sweet)], aiming to identify differentially methylated genes (DMGs) and their relationship with plant drought-responses/adaptation and relevance to cassava plant breeding programs.

## Methods

### Biological assay and data acquisition

**I. Plant materials, water deficit stress treatment.** The experimental design was completely randomized with three replicates, carried out under greenhouse conditions at

Embrapa Mandioca e Fruticultura (12˚40'47.8"S 39˚05'20.7"W) comprising two contrasting cassava varieties (the bitter BRS Formosa and the sweet BRS Dourada), two treatments [non-stressed plants (negative control) and stressed plants (water deficit)] and plots with five plants per variety. BRS Dourada is a local variety with an unknown pedigree, specifically developed for fresh consumption. Its roots exhibit yellow color, and it contains less than 100 ppm of cyanogenic compounds, making it ideal for sweet cassava usages. In contrast, BRS Formosa is a hybrid of industrial cassava, known for its higher concentration of cyanogenic compounds (above 100 ppm), which imparts a bitter taste. The female parent of BRS Formosa is the BGM-0361 accession, while the male parent is unknown. These two varieties display significant genetic differences, as confirmed by molecular analysis through genotyping-by-sequencing techniques [17].

The average temperature inside the greenhouse was regulated by airflow and maintained at maximum temperature at 28˚C ± 3˚C and minimum at 23˚C± 1, with average air relative humidity of 80%. To conduct the experiment, 30-liter plastic pots were utilized filled with a substrate blend consisting of 70% yellow latosol, 20% coconut fiber, and 10% vermiculite. Slow-release fertilization was implemented using 25 grams of specially formulated fertilizer. This fertilizer was encapsulated with elemental sulfur and coated with non-water-soluble organic polymers, containing 11% nitrogen (N), 22% phosphorus pentoxide ($P_2O_5$), 11% potassium oxide ($K_2O$), 12.29% sulfur ($SO_4$), 0.35% boron (B), 0.30% copper (Cu), and 0.30% zinc (Zn).

Throughout the preliminary phase leading up to the water deficit imposition, samples from all treatments were regularly irrigated in order to maintain soil moisture levels close to field capacity. In order to prevent water evaporation from the soil the pots were sealed with aluminum foil. The stress treatment consisted of suspension of irrigation of plants after 50 days after planting. Plants of the negative control maintained the irrigation during the assay, keeping the soil moisture content close to field capacity. The soil moisture content was monitored daily (at 6 am and 6 pm) using a TDR device (Time Domain Reflectometry), with water replacement of the control plants carried out up to field capacity (FC), based on TDR readings. The Fraction of Transpirable Soil Water (FTSW) was calculated by the methodology according to Sinclair and Ludlow (1986) [18]: $FTSW = \frac{\theta day - \theta Final}{\theta initial - \theta Final}$. Leaf gas exchange analysis was carried out in mature leaves – third leaf, between 9:00-10:00am, throughout the experiment using a portable photosynthesis meter ADC BioScientific (model LCpro-SD, Ltd., UK).

When plants presented wilted leaves and with 10% soil moisture (40% of FC), root samples were collected, frozen in liquid nitrogen, and stored in an ultrafreezer until total DNA extraction (CTAB methodology [19]). Total DNA quality and quantity was checked in 1% agarose gel electrophoresis (w/v) and spectrophotometer analysis (NanoVue Plus, GE Healthcare, Waukesha, WI, United States).

**II. MethylRAD-Seq libraries.**   After DNA quantification, the samples (two cassava varieties, two treatments and three replicates – composed by a pool of five plants per variety) were prepared for the construction of twelve MethylRAD-Seq libraries (CD Genomics, USA).

The quality of the raw reads considered the Phred values of the bases, where the filtration step (PEAR software v0.9.6) excluded reads showing more than 15% of bases with a Phred value < 30 or with N content greater than 8% [20]. Reliable methylation sites (CCGG and CCNGG sites) considered sequencing depths greater than 3. The analysis excluded reads without the expected restriction site and Enzyme Reads (those showing unexpected size) of each library. The predicted size tags containing the reference sequences CCGG (32 bp) and CCNGG (31 bp) were mapped (SOAP software v.2.21; parameters: -M 4-v 2 -r 0) in the reference genome of *M. esculenta* v6.1 (Phytozome, https://phytozome-next.jgi.doe.gov/; NCBI RefSeq assembly GCF_001659605.1) disregarding the repeatedly mapped reads.

**III. Mapping of reads in the reference genome.**    The pair-end reads were aligned in the reference genome of *M. esculenta* v6.1 (Phytozome), using BWA aligner v0.7.17-r1188 (BWA-MEM) [21] with the standard parameters. The Samtools v1.7 tool [22] was used to convert SAM files into BAM, draw, remove duplicates, and index data. Next, Picard Tools [23] was used to assign all readings a single new reading group and finally, GATK v4 [24] was used to perform the call for variants to obtain the VCFs files containing the information of the variants by genome.

## Molecular analysis of BRS Formosa and BRS Dourada

**I. Filtering of variants (SNPs/INDELs) and annotation.**    The filtering of the variants was done by the bcftools tool [25] with the parameters QUAL>30 and MQ>30 to generate the filtered VCFs files. For the annotation of the variants regarding the region and the prediction of the effects (impact), the SNPeff [26] was used and the database was created for the reference genome used in the assemblies.

**II. Distribution of methylation sites in structural and functional gene elements.**    Identification of structural gene elements of *M. esculenta*, including introns, exons, UTRs, and regions (2,000 bp) upstream of the TSS (Transcription Start Site), considered the cassava Phytozome dataset (version 12.1; https://phytozome.jgi.doe.gov/pz/portal.html, ID: LTYI01000000). The methylation sites covering the UTRs were counted performing the SNPeff software [26]; and for the other genetic elements, the bedtools software (v2.25.0 https://bedtools.readthedocs.io/en /latest/), was used.

**III. Relative quantification of DNA methylation and analysis of the distribution of methylation sites in gene regions.**    The methylation levels (CCGG and CCNGG target sites) reflected the consistency of the amplification efficiency of isometric sequences and the sequencing depth of the methylated tags. Thus, the methylation level, expressed in RPMs (reads per million), considered the number of reads covering the site by the total of good quality reads in the library × 1,000,000. The RPM calculations also considered three biological replicates (R1, R2, and R3) for each variety and each treatment.

The analysis covering the distribution of methylation sites in genes comprised regions of CDS and 2 kb upstream and downstream of the TSS (Transcription Start Site), and 2 kb upstream and downstream of the TTS (Transcription Terminator Site). The methylation level of two samples (libraries) were analyzed by Pearson's correlation coefficient. Scatterplots of sequencing depths of all methylation sites concerning the pairs of comparisons allowed to verify the consistency of the data of the biological replicates, since values close to 1.00 reflected the best correlations between the two sets of data [27].

**IV. Differential methylation profiles.**    The methylation levels of each cassava variety, when comparing the treatments with and without stress, allowed us to classify those differentially methylated sites as hypermethylated or hypomethylated by performing the edgeR software. The adjusted p-values considered the FDR (False Discovery Rate) technique applied to correct type I errors. The methylation levels comparing the two contrasts [FST (BRS Formosa Stressed Treatment) vs FNC (BRS Formosa Negative Control) and DST (BRS Dourada Stressed Treatment) vs DNC (BRS Dourada Negative Control)] considered for each treatment a pooled sample composed by the three biological replicates showing sequencing depths greater than three. Differently methylated sites were associated with previously annotated genes, whose expressions ($Log_2$ RPM values) were illustrated in heatmaps generated by the Tbtools v1.09 software. Venn diagrams (https://bioinfogp.cnb.csic.es/tools/venny/) contemplating the sets of differentially methylated sites identified those common or exclusive identifiers from the BRS Formosa or BRS Dourada drought-stress responses.

**V. Analysis of gene ontology and prediction of interactomes.** The sets of genes showing differential methylation levels (hyper and hypomethylated for CCGG and CCNGG sites) in structural gene elements of BRS Formosa and BRS Dourada were characterized in terms of gene ontology (GO), applying a PlantRegMap tool (http://plantregmap.cbi.pku.edu.cn/). The analysis also identified those enriched terms (Fisher's tests, p-value < 0.01), comparing the input sets with the global *M. esculenta* data available.

Proteins encoded by hyper- or hypomethylated genes of each variety helped us to predict potential protein-protein interaction (PPI) networks performing the STRING tool (v11.0b; https://string-db.org/), using *A. thaliana* orthologs as a reference model. The interactions showed a minimum score of 0.7 (high confidence) and maximum interactors of 20 (1st shell) and 10 (2nd shell). Also, only interactions derived from experimental data or co-expression/co-occurrence of results were reported. A functional enrichment analysis using the Aggregate Fold Change method [28] identified the main biological processes, molecular functions, and cellular components concerning the PPI members. Additional information involved hierarchical local clusters, metabolic pathways, and annotations derived from the STRING analysis.

## Results

Soil water content played a significant role in shaping the dynamics of plant gas exchange throughout the imposition of water deficit conditions. Regarding genotypic responses to soil water availability, the BRS Formosa variety exhibited a significantly higher water extraction capacity compared to BRS Dourada. BRS Formosa depleted the transpirable soil water fraction more rapidly at the severe stress level, which explains the pronounced impact on plant gas exchange throughout the study, particularly when compared to BRS Dourada; which somewhat reflected in variations in DNA methylation. These differences are readily detected when comparing the gas exchange of genotypes subjected to three progressively increasing levels of water deficit ($WD_1$, $WD_2$, and $WD_3$) over a 15-day highlighted period of soil drying. In $WD_3$, the soil water content values were 0.12 $m^3 m^{-3}$ for BRS Formosa and 0.5 $m^3 m^{-3}$ for BRS Dourada (S1–S3 Figs).

The soil water deficit significantly affected the gas exchange of plants (gs - stomatal conductance to water vapor, $mol.m^{-2}.s^{-1}$; E - leaf transpiration rate, $mmol.m^{-2}.s^{-1}$; and A - net photosynthetic rates, $\mu mol.m^{-2}.s^{-1}$), with progressively decreasing values for gs, E, and A, and a gradual increase in water use efficiency A/E. The reductions in gas exchange were more pronounced for the BRS Formosa variety, particularly in terms of transpiration. Quantifying by considering the values of the normalized transpiration rate (Es/Ec - 1); in WD3, BRS Dourada maintained transpiration levels at 67% compared to control plants, while BRS Formosa was at 25%. These values resulted in higher water use efficiency (A/E) for the BR Formosa variety influenced by stomatal regulation. Interactions between the factors were observed primarily for A, with the most pronounced effects of water deficit ($WD_3$) significantly affecting gas exchange in the BRS Formosa variety (S1 and S2 Figs).

An integrated analysis of the soil-water-plant relationships using the complete dataset (gas exchange and soil water content) during the water deficit period led to a reduction in the ratio between the normalized transpiration rate (Es/Ec) and the Fraction of Transpirable Soil Water (FTSW = $m^3.m^{-3}$ / $m^3.m^{-3}$), as shown in Fig 1. Furthermore, distinct variations in physiological sensitivity and responses to water deficit were evident among the different genotypes based on these soil-plant interactions. Notably, differences in the inflection points of the models became apparent, as the transpiration rate remained constant up to 0.50 of FTSW for the BRS Formosa variety and 0.35 of FTSW for the BRS Dourada variety. These response patterns underscored the contrasting behaviors of these cultivars in terms of their sensitivity to water deficit stress and, consequently, their production of endogenous compounds. These physiological responses

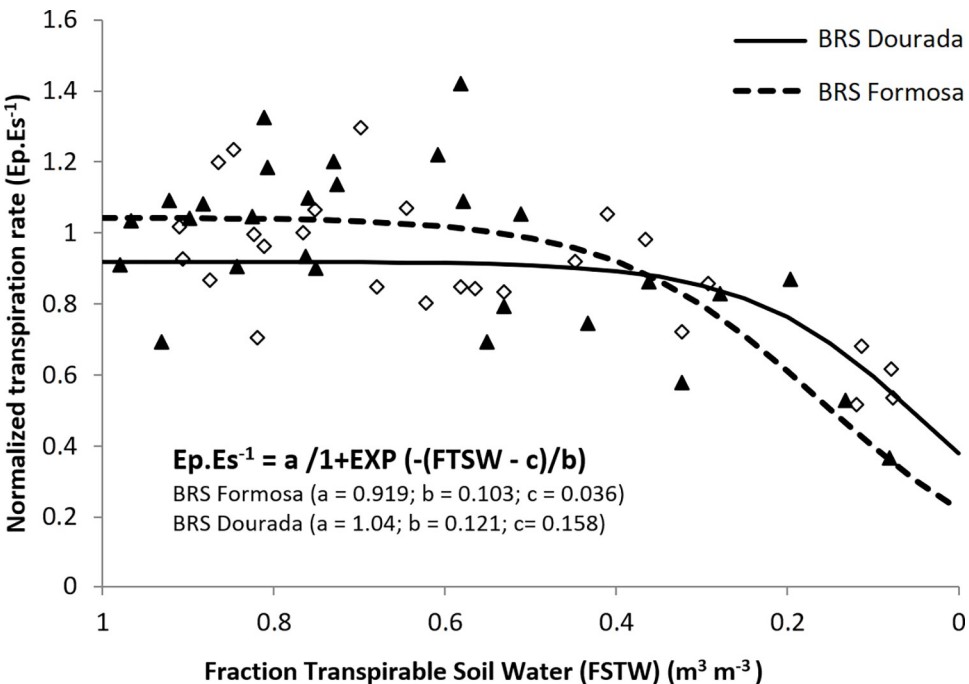

**Fig 1. Relationship between the normalized transpiration rate (Es.Ec⁻¹) and the fraction of transpirable soil water during the investigation of the soil-drying period.** The lines represent the fitted non-linear model for the pooled data sets of each cassava genotype. Where θ is the soil water content ($m^3.m^{-3}$).

were particularly evident in the pronounced wilting observed in cassava leaves subjected to severe stress. Detailed images of BRS Formosa and BRS Dourada plants under both controlled and water-stress conditions are presented in Fig 2.

## Sequenced MethylRAD data

The sequenced MethylRAD data (Table 1) revealing the tags with methylated sites when mapped in the *M. esculenta* genome, covered from 17.90 to 21.24%, depending on the variety and the treatment. The total methylation sites (CCGG or CCNGG) and the respective mean depths of sequencing libraries are shown in Table 2. For the CCGG sites, the mean depths ranged from 15.67 (FNC-R3) to 21.25% (FNC-R1) and for the CCNGG sites, these values ranged from 11.56 (DNC-R2) to 14.77% (FNC-R2). All libraries (treatments) presented sequencing depths higher than three (the minimum considered in the analysis).

## Correlation between methylation levels between samples

The methylation levels of CCGG and CCNGG sites detected in each library reflected the sequencing depths. Scatter plots covering the sequencing depths of two libraries, according to their Pearson correlation coefficients (S7 and S8 Figs), were consistent between the biological replicates of each variety. About the biological replicates of BRS Dourada, considering the methylated CCGG site, the data was consistent, showing r ≥ 0.86 (non-stressed treatment) and r ≥ 0.92 (stressed treatment). As to the methylated CCNGG site, the data was also consistent, showing r ≥ 0.71 (non-stressed treatment) and r ≥ 0.82 (stressed treatment). the data of the biological replicates of BRS Formosa and the methylated CCGG site, were also consistent, presenting r ≥ 0.94 (non-stressed treatment) and r ≥ 0.85 (stressed treatment). The data of the methylated CCNGG site, was also consistent, with r ≥ 0.96 (non-stressed treatment) and

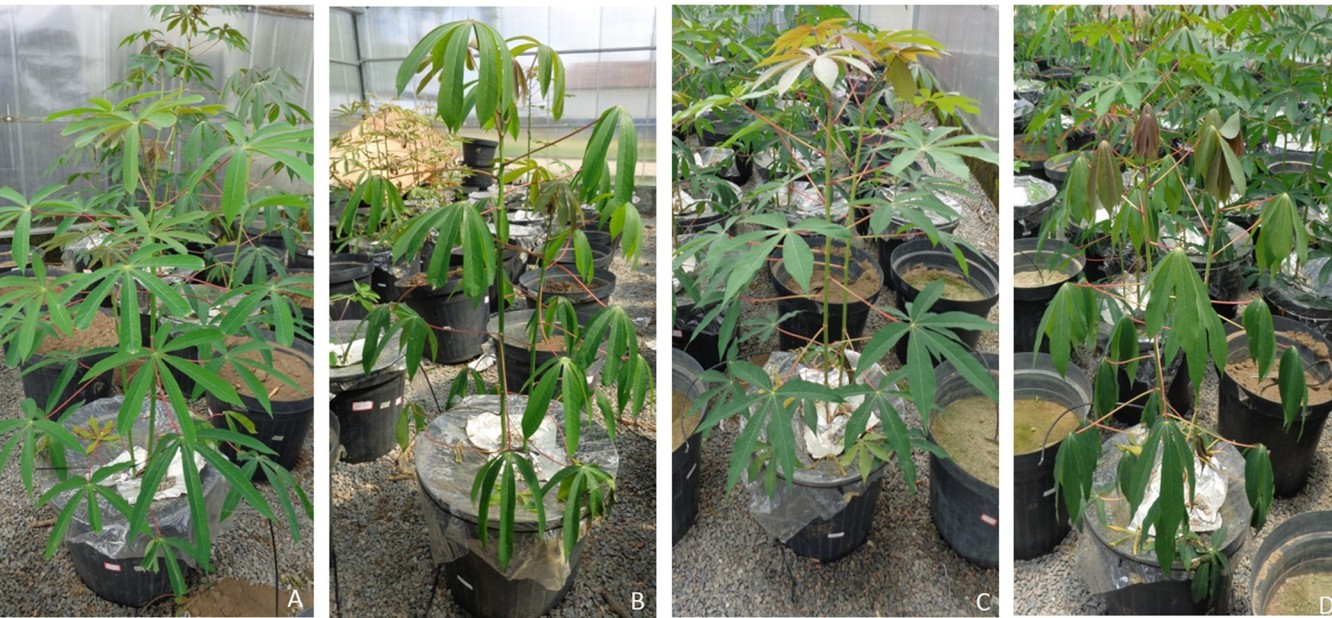

**Fig 2.** A) Non-stressed BRS Formosa plants with soil water content of 0.20 m³.m³ (soil at field capacity); B) BRS Formosa plants under visual severe stress with soil water content at 0.10 m³.m³; C) Non-stressed BRS Dourada plants with soil water content of 0.20 m³.m³ (soil at field capacity); D) BRS Dourada plants under visual severe stress with soil water content at 0.10 m³.m³.

r $\geq$ 0.88 (stressed treatment). Therefore, the biological replicates of the two varieties were consistent in both studied conditions, enabling comparisons.

## The methylated sites in structural genetic elements

The distribution of methylated CCGG and CCNGG sites detected in structural gene elements is shown in Fig 3. The CCGG methylated sites expressively occurred mainly in intergenic,

**Table 1. Total MethylRAD sequencing data in the *Manihot esculenta* Crantz reference genome v6.1 Phytozome.**

| Sample | Enzyme Reads | Mapping Reads | Proportion | FPKM |
|---|---|---|---|---|
| A01-FST R1 | 8,473,392 | 1,516,553 | 17.90% | 0.0030607 |
| A02-FST R2 | 7,894,473 | 1,643,074 | 20.82% | 0.0033161 |
| A03-FST R3 | 8,855,136 | 1,667,859 | 18.84% | 0.0033661 |
| A04-FNC R1 | 8,703,725 | 1,764,280 | 20.27% | 0.0035607 |
| A05-FNC R2 | 7,596,265 | 1,525,598 | 20.09% | 0.0030790 |
| A06-FNC R3 | 6,442,813 | 1,240,765 | 19.25% | 0.0025041 |
| A07-DST R1 | 6,467,929 | 1,373,931 | 21.24% | 0.0027729 |
| A08-DST R2 | 7,530,213 | 1,364,509 | 18.12% | 0.0027539 |
| A09-DST R3 | 7,419,768 | 1,434,261 | 19.33% | 0.0028947 |
| A10-DNC R1 | 6,567,305 | 1,220,180 | 18.58% | 0.0024626 |
| A11-DNC R2 | 5,902,892 | 1,222,560 | 20.71% | 0.0024674 |
| A12-DNC R3 | 5,095,637 | 1,056,007 | 20.72% | 0.0021312 |

Total MethylRAD sequencing data of expected size reads (Enzyme Reads with 31-32 bp) with a single mapping site in the *Manihot esculenta* reference genome (v6.1 Phytozome) of 12 samples/treatments and quantities mapped in the genome (Mapping Reads), their proportions (%) and FPKM values (multiplied by 106). Samples/ treatments: FST (BRS Formosa, Stressed Treatment); FNC (BRS Formosa, Negative Control); DST (BRS Dourada, Stressed Treatment); DNC (BRS Dourada, Negative Control); R1, R2 and R3: biological triplicates.

**Table 2. Methylated sites for each MethylRAD library.**

| Sample | CCGG | | CCNGG | |
|---|---|---|---|---|
| | Number of sites | Deep | Number of sites | Deep |
| A01-FST-R1 | 40,104 | 17.26 | 33,321 | 14.10 |
| A02-FST-R2 | 41,467 | 20.7 | 28,055 | 14.59 |
| A03-FST-R3 | 42,306 | 17.55 | 34,667 | 15.63 |
| A04-FNC-R1 | 43,275 | 21.25 | 30,921 | 14.43 |
| A05-FNC-R2 | 38,463 | 18.1 | 32,124 | 14.77 |
| A06-FNC-R3 | 35,957 | 15.67 | 29,918 | 12.45 |
| A07-DST-R1 | 36,555 | 19.66 | 25,051 | 13.13 |
| A08-DST-R2 | 36,433 | 16.35 | 31,662 | 13.83 |
| A09-DST-R3 | 36,084 | 18.17 | 30,887 | 14.16 |
| A10-DNC-R1 | 31,094 | 17.47 | 28,279 | 13.39 |
| A11-DNC-R2 | 34,724 | 17.44 | 27,314 | 11.56 |
| A12-DNC-R3 | 29,323 | 17.69 | 21,228 | 12.73 |

Total methylated sites (with sequencing depth greater than 3), and the average sequencing depth for each MethylRAD library. Treatments described: FST (BRS Formosa, Stressed Treatment); FNC (BRS Formosa, Negative Control); DST (BRS Dourada, Stressed Treatment); DNC (BRS Dourada, Negative Control); R1, R2 and R3: biological triplicate

exons, introns, and defined upstream regions (Fig 3A). For the CCNGG sites, most detections occurred in intergenic, upstream, introns and exons regions (Fig 3B). In both cases, the methylated sites in the UTRs, were less detected.

Regarding the genes, the highest levels of methylated sites (CCGG and CCNGG), based on RPM values, comprised the initial 20% of the gene lengths (S4 Fig). Also, the methylated sites (CCGG and CCNGG) occurred around the TSS (S5 Fig). An additional peak concerning the CCGG site occurred around 1,500 bp downstream to the TTS. Another methylation peak (CCGG and CCNGG) occurred about 500 bp upstream to the TTS (S6 Fig).

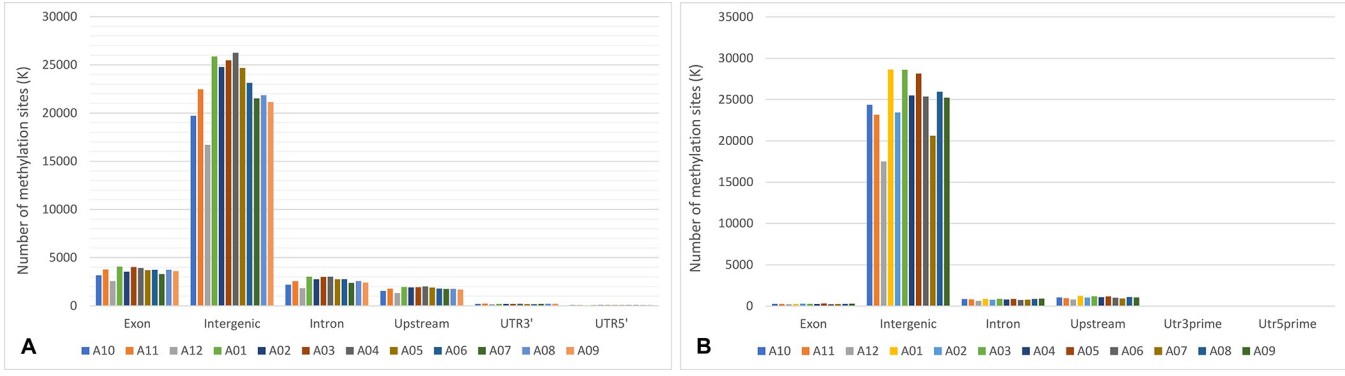

**Fig 3. a)** Total CCGG-type methylated sites in different gene regions of the *M. esculenta* genome [upstream: region 2000 bp prior to the start of transcription]. A01: FST-R1; A02: FST-R2; A03: FST-R3; A04: FNC-R1; A05: FNC-R2; A06: FNC-R3; A07: DST-R1; A08: DST-R2; A09: DST-R3; A10: DNC-R1; A11: DNC-R2; A12: DNC-R3; FST (BRS Formosa, Stress Treatment); FNC (BRS Formosa, negative control); DST (BRS Dourada, Stress Treatment); DNC (BRS Dourada, negative control) R1, R2 and R3: biological triplicates. **b)** Total CCNGG-type methylated sites in different gene regions of the *M. esculenta* genome [upstream: region of 2000 bp prior to the start of transcription]. A01: FST-R1; A02: FST-R2; A03: FST-R3; A04: FNC-R1; A05: FNC-R2; A06: FNC-R3; A07: DST-R1; A08: DST-R2; A09: DST-R3; A10: DNC-R1; A11: DNC-R2; A12: DNC-R3; FST (BRS Formosa, Stress Treatment); FNC (BRS Formosa, negative control); STD (BRS Dourada, Stress Treatment); DNC (BRS Dourada, negative control); R1, R2 and R3: biological triplicates.

**Table 3. Methylated sites in the genome of *Manihot esculenta*.**

| Contrast | Site | Total | Genes | Methylation | |
|---|---|---|---|---|---|
| | | | | Hyper | Hypo |
| BRS Formosa (FST *vs* FNC) | CCGG | 775 | 63 | 42 | 21 |
| | CCNGG | 807 | 14 | 9 | 5 |
| Subtotal | | 1582 | 77 | 51 | 26 |
| BRS Dourada (DST *vs* DNC) | CCGG | 858 | 89 | 55 | 34 |
| | CCNGG | 783 | 24 | 14 | 10 |
| Subtotal | | 1641 | 113 | 69 | 44 |
| Total | | 3223 | 190 | 120 | 70 |

Total methylated CCGG/CCNGG sites in the genome of *Manihot esculenta* and differentially methylated (hyper and hypomethylated) genes of BRS Formosa and BRS Dourada varieties when comparing the respective treatments with and without water deficit. Sample/treatment: FST (BRS Formosa, Stressed Treatment); FNC (BRS Formosa, Negative Control); DST (BRS Dourada, Stressed Treatment); DNC (BRS Dourada, Negative Control).

## The differentially methylated sites

The totals of methylated sites (CCGG and CCNGG) and the DMGs (classified as hyper or hypomethylated), considering the contrast stressed *versus* non-stressed of each variety, are shown in Table 3. The sweet BRS Dourada showed more methylated sites (and DMGs). Consequently, also more hyper and hypomethylated genes than the bitter BRS Formosa.

Heatmaps showing the profiles (each methylated site) of DMGs in BRS Dourada and BRS Formosa considering their replicates (R1, R2, and R3) after stress or without stress, are shown in S9 Fig (BRS Formosa) and S10 Fig (BRS Dourada). These heatmaps reinforce the data consistency of the replicates of each variety reflecting the Pearson correlation coefficients. Annotations of the DMGs are provided in S1 Table.

The analysis of *M. esculenta* identifiers (Manes from cassava genome v6.1, Phytozome 12.1) associated with the differentially methylated sites (Fig 4A) showed most of them exclusive to one or another variety, only sharing 12.4% of these identifiers. The same pattern was observed with the identifiers associated to genes (Fig 4B). A Venn diagram comparing the sets of hyper and hypomethylated genes of each variety after stress (Fig 4C) highlights the genes detected in both variety's profiles (only one hypo and three hypermethylated genes). The results stood out the distinct DNA methylation profile of each accession, reinforcing the analysis of polymorphisms covering the DMGs of both varieties.

## Polymorphisms covering differentially methylated genes

The analysis of polymorphisms (SNP/INDEL) considering the DMGs, when comparing the genomic sequences of each variety with the reference *M. esculenta* v.6.1, a total of 15,643 base changes highlighted 168 genes (of the 190 detected DMGs in both varieties). Of these, 13,196 alterations could be classified as modifiers with no major effects on gene information, while 2,447 changes were considered low (1,340), moderate (1,042) or of high (65) impact. Changes considered modifiers included events of: UTR3' variant; variant intron; UTR5' variant; downstream gene variant. Events of low impact involved: UTR5' premature start codon gain variant; splice region variant; splice region variant and intron variant; synonymous variant. Effects of moderate impact included: missense variant; conservative in-frame deletion; disruptive in-frame deletion; disruptive in-frame insertion; missense variant/splice region variant. In contrast, high-impact effects involved: frameshift variant; frameshift variant/splice region variant; frameshift variant/start lost; frameshift variant/stop lost; splice acceptor variant/intron variant; splice acceptor variant/splice region variant/UTR3' variant/intron variant; splice donor

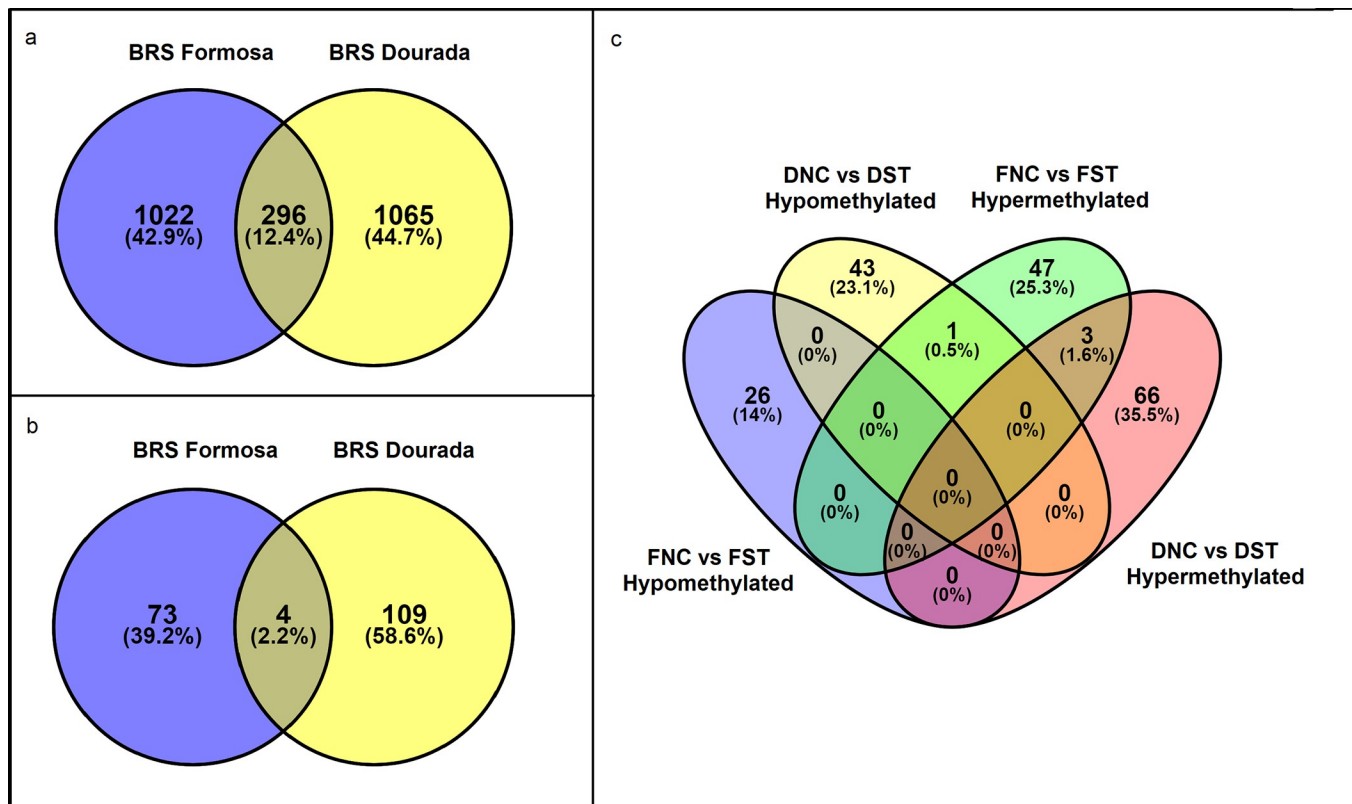

**Fig 4. Venn diagram using Manes identifiers (*Manihot esculenta*, Phytozome v.12.1) with differential methylation sites detected for BRS Formosa (bitter) and BRS Dourada (sweet) varieties in comparison to contrast with and without water deficit. a)** all identifiers detected with methylated sites (CCGG and CCNGG); **b)** methylated identifiers related to genes; **c)** Venn diagram using Manes identifiers (*M. esculenta*, Phytozome v.12.1) with hyper and hypomethylated sites (CCGG and CCNGG) in genes, detected in the contrast comparison with and without water deficit for the biiter variety BRS Formosa (FNC vs FST) and for the sweet BRS Dourada (DNC vs DST); FST (BRS Formosa, Stress Treatment); FNC (BRS Formosa, negative control); DST (BRS Dourada, Stress Treatment); DNC (BRS Dourada, negative control).

variant/UTR5' variant/intron variant; splice donor variant/intron variant; splice donor variant/splice region variant/UTR5' variant/intron variant; stop gained; stop lost; stop lost/splice region variant. The totalization of low, moderate and high impact events detected by candidate gene is given in S2 Table, as well as, for instance, the position in the genome of one of the most impacted events, the reference base in the genome v.6.1 and the altered nt(s) in addition to the homozygous or heterozygous condition of polymorphism if detected in each variety and the status (hyper or hypomethylation).

## Enriched GO terms and the predicted interactomes

The enriched gene ontology (GO) terms associated with DMGs of each variety is provided in S3 Table. For the bitter BRS Formosa, 41 of 51 hypermethylated genes (both sites) highlighted enriched GO terms related with peptidase and proteolysis. In turn, enriched GO terms related with 20 of the 26 hypomethylated genes highlighted to the cell cycle regulation and plant responses [to ozone, ROS, and jasmonic acid (JA)]. Regarding the sweet BRS Dourada, enriched GO terms relative to 56 of 69 hypermethylated genes highlighted the Golgi complex and endosomes, while enriched GO terms related to 32 of the 44 hypomethylated genes highlighted the DNA repair, also transport and channel activities, and regulations (cell cycle, organelles, and DNA).

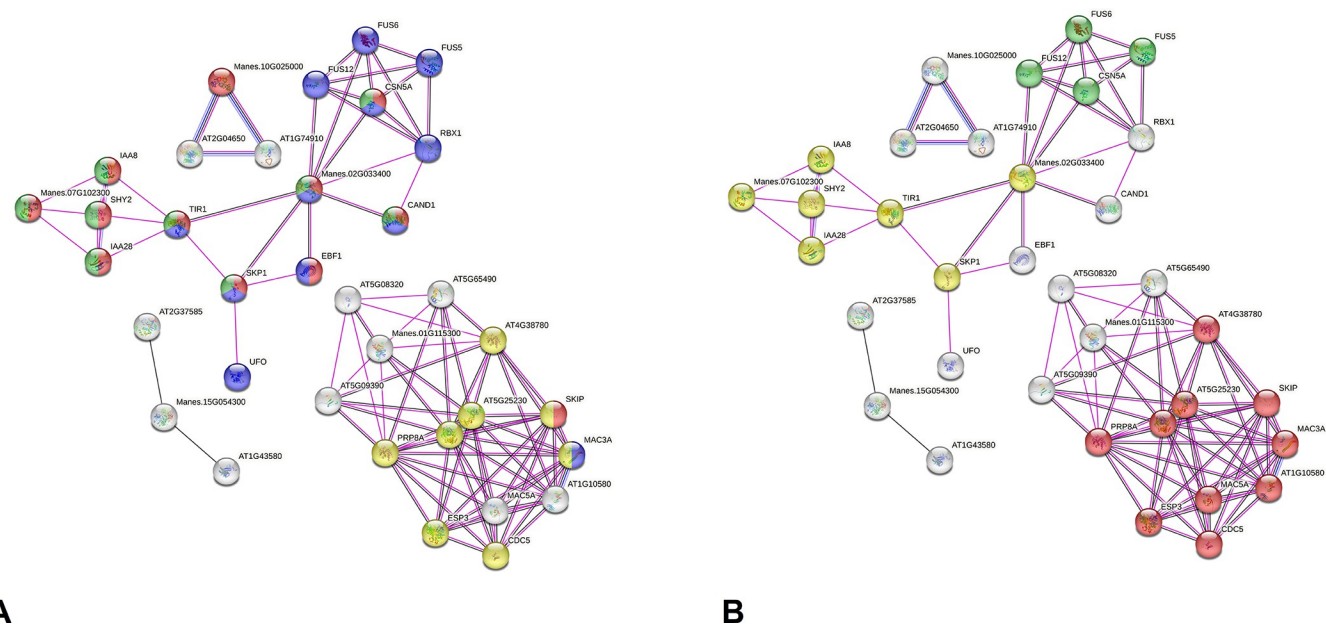

**A** **B**

**Fig 5. Interactome predicted for hypomethylated gene products (CCGG and CCNGG sites) in the BRS Formosa variety *(M. esculenta)* after water deficit stimuli based on *A. thaliana* ortholog proteins and the STRING tool.** The spheres represent proteins and the edges the interactions (pink: experimental; black: co-expression; blue: co-occurrence). Input file proteins are identified by the prefix "Manes". **a)** Sphere colors represent GO terms for biological processes (Dark blue: modification of proteins by conjugation or removal; light green: auxin signaling activation pathways; yellow: RNA splicing; red: hormonal response); **b)** Colors of the spheres represent Keywords (Yellow – auxin signaling pathways), KEGG pathways (red – Spliceosome), local cluster (green – Sinalosome).

The PPI network predicted with proteins of hypomethylated (both sites) genes of the bitter BRS Formosa after the stress stood out five of 26 ortholog proteins (based on *the A. thaliana* model) (Fig 5). This network (56 nodes and 95 edges; average of 3.39 nodes) presented an average local clustering coefficient of 0.495. The GO terms derived from the PPI members pointed 28 enriched biological process terms, five of them related to stimuli and hormonal responses, especially to auxin. The analysis detected a cluster for COP9 signalosome with interactions highlighting the auxin signaling (Fig 5B). From six enriched molecular function terms, the highlighted was pre-mRNA processing (S4 Table, Fig 5A). From proteins without predicted interactions five presented GO term response to stimuli (GO:0050896). Proteins also reinforced the spliceosome participation (S4 Table).

The predicted PPI network derived from eight hypermethylated (both sites) genes of the bitter BRS Formosa highlighted five of 47 *M. esculenta* orthologs. This network (77 nodes and 180 edges; average of 4.68 nodes) presented an average local clustering coefficient of 0.444 (Fig 6). As to the 68 enriched GO terms for biological process, they highlighted mainly primary metabolic processes (GO:0044238), gene expression regulation (GO:0010468), and proteolysis (GO:0006508) (S5 Table and Fig 6A). Proteins without interactions also highlighted such enriched terms. Regarding the 13 enriched GO terms for molecular function, transcription factor binding, was well represented. Additional enrichment exploring the KEGG data (S5 Table and Fig 6B) highlighted 15 proteins interacting with basal transcription factors (TF), 12 with proteasomes, and five with RNA degradation.

In turn, the PPI network derived of hypomethylated (both sites) genes of the sweet BRS Dourada highlighted six of 42 *M. esculenta* orthologs. The network (72 nodes and 250 edges; an average of 6.94 nodes) presented an average local clustering coefficient of 0.453 (Fig 7). The

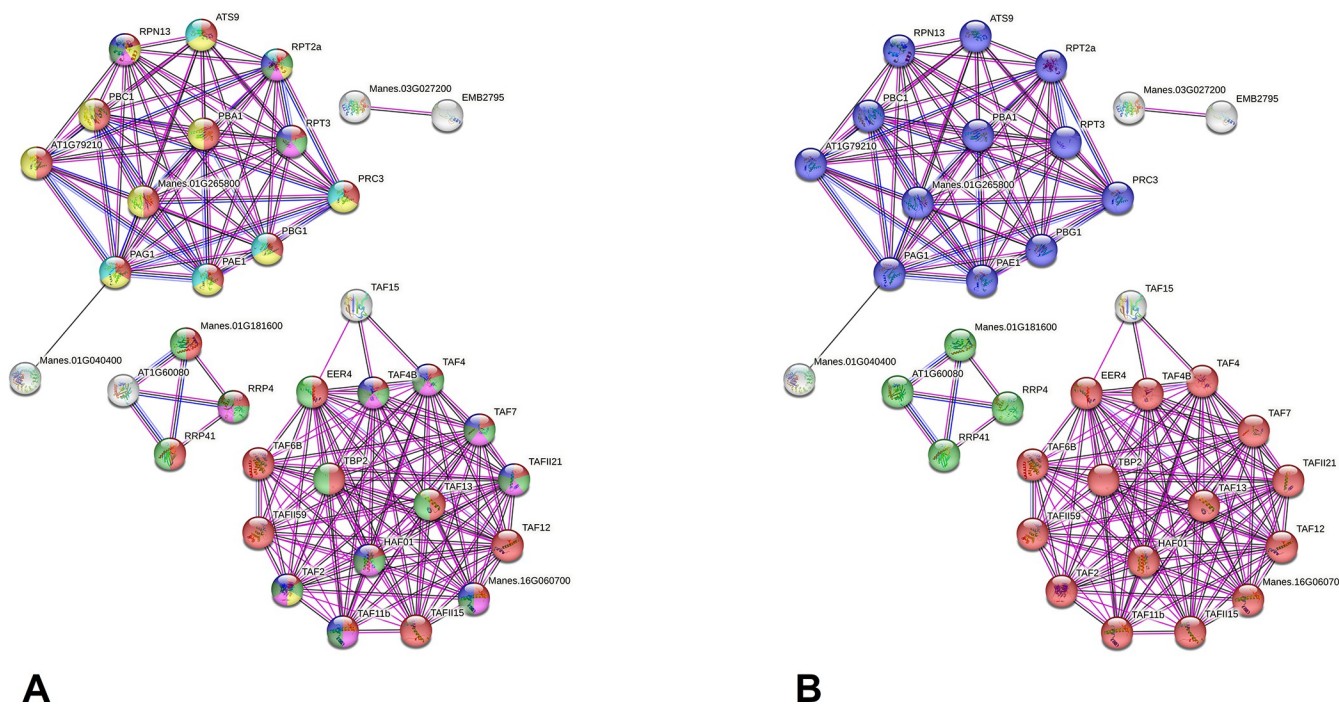

**Fig 6. Interactome predicted for hypermethylated gene products (CCGG and CCNGG sites) of the BRS Formosa variety (*M. esculenta*) after a water deficit stimulus, according to *A. thaliana* model and STRING tool.** The spheres represent proteins and edges the interactions (in pink, the experimental ones; in black, those of co-expression; in blue, those of co-occurrence). Input file proteins are prefixed with "Manes". **a)** Colors of the spheres represent GO terms for biological processes (red- primary metabolic process; light green: regulation of gene expression; yellow: proteolysis; lilac: up-regulation of biological processes; dark green: up-regulation of metabolic processes of macromolecules; dark blue: upregulation of the metabolic process of nitrogen compounds; light blue: response to metal ions); **b)** Colors of the spheres represent highlighted terms from the KEGG database (blue: proteasome; green: RNA degradation; red: basal transcription factors).

GO characterization revealed 36 enriched biological process terms with emphasis on primary metabolic processes (GO:0044238) and gene expression (GO:0010467), in addition to 16 for molecular function (S6 Table). Proteins without predicted interactions also highlighted enriched GO terms, specially related with gene expression and primary metabolic process (Fig 7A). Additionally, the enrichment exploring the KEGG data showed 13 proteins related to ribosomes, five specifically to ribosome biogenesis, and six to mismatch repair of genetic material (S6 Table and Fig 7B).

A supposed PPI network involving proteins of the hypermethylated (both sites) genes of the sweet BRS Dourada highlighted 11 of 65 *M. esculenta* orthologs. This network (95 nodes and 182 edges; an average of 3.83 nodes) presented an average local clustering coefficient of 0.402 (Fig 8). The GO characterization revealed 56 enriched terms (biological process) highlighting primary metabolic process (GO:0044238), organization of the cellular component (GO:0016043), DNA replication (GO:006260), and stress response (GO:0033554), besides 16 enriched terms for molecular function (S7 Table, Fig 8A). Proteins without interactions also highlighted enriched terms related with primary metabolic process, cellular component organization, and stress responses (S7 Table). The enrichment exploring the KEGG data highlighted proteins related to DNA replication and degradation, and RNA transport (S7 Table). Also, clusters highlighted DNA replication, DNA polymerase complex, exoribonuclease complex, initiation factor, and cell death programming (Fig 8B).

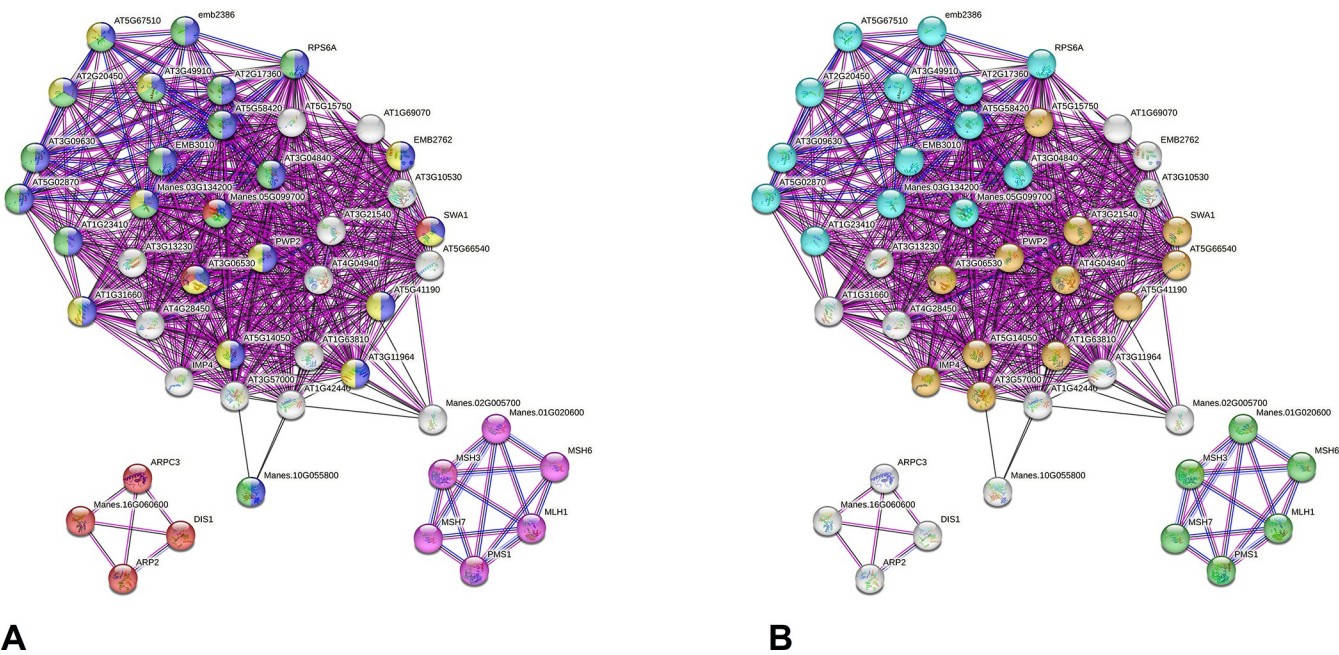

**Fig 7. Interactome predicted for hypomethylated gene products (CCGG and CCNGG sites) of the BRS Dourada variety (*M. esculenta*) after a water deficit stimulus, according to *A. thaliana* model and STRING tool.** The spheres represent proteins and edges the interactions (pink: the experimental ones; black: those of co-expression; blue: those of co-occurrence). Input file proteins are prefixed with "Manes". **a)** The colors in each node represent GO terms for biological processes (blue: gene expression; light green: translation; yellow: ribosomal biogenesis; red: regulation of the cellular process; purple: Mismatch repair); **b)** Colors at each node represent KEGG terms (Orange: Ribosomal biogenesis in eukaryotes; green: Mismatch repair; light blue: ribosome).

## Discussion

### DNA methylation profiles of the cassava varieties and the methylated sites in structural genetic elements of *M. esculenta*

The DNA methylation profiles of the bitter BRS Formosa and the sweet BRS Dourada were somewhat distinct considering their stress responses. These methylome profiles were reinforced by the individual profiles of the replicates (R1, R2, and R3), as shown by the adequate Pearson correlation coefficients.

Most DNA methylated sites (CCGG and CCNGG) occurred in intergenic regions of the *M. esculenta* genome, in agreement with the regular detection in eukaryotic genomes [29]. About structural elements of genes, CCGG-methylated sites occurred more in exons, introns, and upstream regions of TSSs. In turn, methylated CCNGG sites occurred more in introns and upstream regions (TSS) than in exonic regions. In plants, methylated sites were distributed mainly in introns and upstream regions [30]. Thus, both cassava varieties presented comprehensive DNA methylation patterns, being exons well represented for at least the CCGG sites, the most methylated site. Brenet *et al.* [31] reported that DNA methylation downstream of the TSS, in the first exon region is much more strongly linked to transcriptional silencing than methylation in the promoter region upstream of the TSS. Thus, such CCGG methylated sites have the potential to influence the regulation of *M. esculenta* genes in response to drought stress.

Since most of the methylated sites occurred in the initial 20% of the gene lengths, close to the TSS, this detection reinforces those previously reported in the literature, which emphasizes the methylated sites close to the TSS and along exons and introns of angiosperms [32]. In such locations, methylated sites generally affect gene expression since they can inhibit RNA

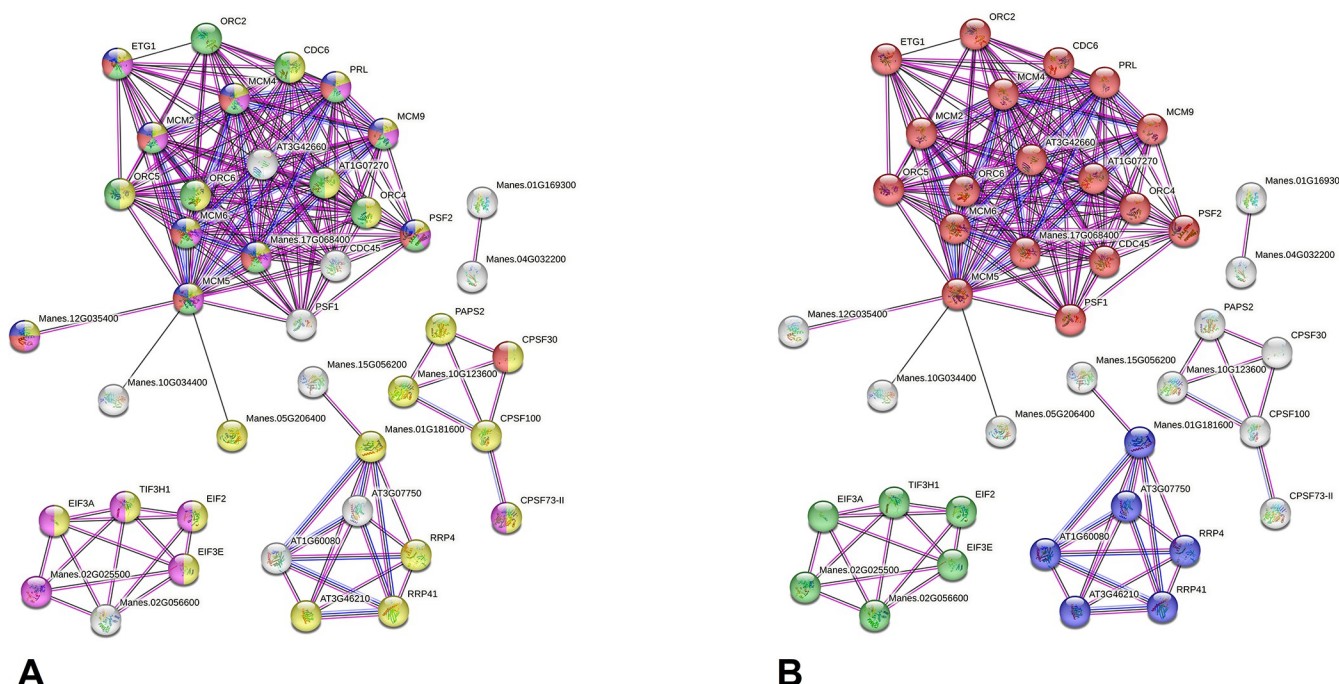

**Fig 8. Interactome predicted for hypermethylated gene products (CCGG and CCNGG sites) of the BRS Dourada variety (*M. esculenta*) after a water deficit stimulus according to the *A. thaliana* model and STRING tool.** The spheres represent proteins, and edges, the interactions (pink - the experimental ones; black- those of co-expression; blue- those of co-occurrence). Input file proteins are prefixed with "Manes". **a)** Colors of the spheres represent GO terms for biological processes (yellow: primary metabolic processes; purple: organization of cellular components; light green: DNA replication; red: cellular stress response; dark blue: DNA repair); **b)** Colors of the spheres represent terms for local Cluster (red: DNA replication and DNA Polymerase III complex; dark blue: exoribonuclease complex; green: cell death initiation and programming factor).

polymerase II (Pol II) and transcriptional initiation, which would explain the low expression levels of methylated genes [33]. In contrast, considering the final transcribed region, the methylation peak (both sites) covered the defined region upstream of the transcription termination site (TTS); although a second smaller peak for CCGG sites was detected downstream of the TTS. Bartels *et al*. [14] report that DNA methylation decreases significantly around the TTS, where the effectiveness of transcriptional control is lower than around the TSS regions.

The analysis identifying differentially methylated sites (CCGG/CCNGG) reinforced the distinct methylation profile of the varieties. Both varieties only shared 12% of these sites (four DMGs). Among them are two hypermethylated genes, probably silenced. These genes encode a hydrolase and a nuclease, with possible actions in DNA repair. Such hypermethylation may not compromise the stress-induced response of the bitter BRS Formosa, probably redirecting energy toward the expression of more effective genes. In turn, the hypermethylation could compromise the stress response of the sweet BRS Dourada if the stress-induced DNA damages.

Additional DMG showed hypermethylation in the bitter BRS Formosa and hypomethylation in the sweet BRS Dourada. This DMG encodes a RING/U-box protein acting in protein ubiquitination [34]. The methylated profiles agreed with the varieties' drought responses since ubiquitination leads to protein degradation, being relevant in plants' adaptation to drought. Thus, the sweet BRS Dourada probably activated the degradation of proteins damaged by the stress, while the bitter BRS Formosa did not require such gene induction.

The GO analysis associated with hyper or hypomethylated genes of each variety allows us to identify enriched GO terms. The enriched GO terms concerning hypermethylated genes of the bitter BRS Formosa reflected primary metabolic processes, especially involving enzyme

activities (hydrolase, peptidase, endopeptidase, carboxypeptidase, exopeptidase, metallopeptidase, metalloendopeptidase) and proteolysis. This hypermethylation, possibly silencing these genes, minimizes the basal metabolism, aiming to redirect energy/substrates to express more effective stress-responsive genes. The GO characterization reinforced the expected stress response of a drought-tolerant variety. The enriched GO terms related to the hypomethylated genes of the bitter BRS Formosa highlighted cell cycle regulation, response to reactive oxygen species (ROS), and response to jasmonic acid (JA). This profile agrees with the expected stress response of a drought-tolerant variety. Su *et al*. [35] reported a drought-tolerant plant (*A. thaliana*) inducing genes of the JA biosynthesis pathway after overexpression of the *VaNAC17* (*Vitis amurensis*) transgene. This gene encodes a TF that induces transcription of drought-responsive genes involved in the JA biosynthesis pathway (*lipoxygenase 3*, *allene oxide cyclase 1*, and *oxophytodienoate reductase* 3). Thus, the transgenic plants increased drought tolerance after water deficit (6% PEG 6000; -0.2 MPa) and salinity (100 mM NaCl) stresses by reducing ROS accumulation.

The enriched GO terms involving hypermethylated genes of the sweet BRS Dourada highlighted the trans-Golgi network, the Golgi sub-compartment, and the endosome. These terms are associated with basal cell functions, affecting the traffic of proteins and lipids among the subcellular compartments. Such hypermethylation silencing those genes helps BRS Dourada save energy/substrates, trying a more effective drought-stress response. The enriched GO terms associated with hypomethylated genes of the BRS Dourada pointed to basal cellular functions, highlighting the regulation of anatomical structure size, cell cycle process regulation, and DNA repair. The size of plant structures and cell cycle regulation indicate plant development control during stress, probably reducing plant growth and development, saving energy/substrates to perform a more effective stress response. However, the emphasis on DNA repair may reflect potential macromolecule damages caused by enhanced ROS generation and oxidative stress. Thus, the methylated pattern and the GO characterization agree with the expected stress-response of such drought-sensitive variety after the stress.

### The bitter BRS Formosa's interactome

Regarding the bitter BRS Formosa, while the hypomethylated genes highlighted plant strategies involving plant cell wall regulation, hormonal stress-signaling, and auxin response factors (ARFs), the hypermethylated genes, potentially silenced, contribute to reducing the protein degradation and action of the KPNB1 protein, which gene inactivation already reported showed to increase plant's drought tolerance. Details are provided below.

**I. The PPI networks based on BRS Formosa's hypomethylated genes.** The PPI networks involving proteins encoded by hypomethylated genes of the bitter BRS Formosa after the stress, pointed a mannose-1-phosphate guanylyltransferase 1 (CYT1, Manes.10G025000) interacting with the ADP-glucose pyrophosphorylase protein, an enzyme acting in biosynthetic processes. The CYT1 protein plays a key role in plant cell wall development since *A. thaliana* loss-of-function mutant plants show irregular cell walls, affecting the cell turgor [36]. Turgor pressure is critical to cell wall growth and expansion, and physiological changes in plants under drought showed loss of turgor, with low turgor pressure affecting the adaptation of the plant's hydraulic system, reducing plant growth [37]. Comparing with the reference *M. esculenta* genome (v.6.1), the CYT1 gene presented SNPs/INDELs polymorphisms (six of low impact and two of moderate effects). One of moderate impact (a missense alteration) was in heterozygous condition in BRS Formosa (non-polymorphic in BRS Dourada; S2 Table). This SNP/INDEL deserves to be explored if such condition provides a selective advantage for a drought-tolerant variety. The FASTA sequence is shown in S2 Table.

A potential interaction of cullin-like protein1 (Manes.02G033400) and auxin response factor 6 (ARF6; Manes.07G102300) highlighted the plant's hormonal response. Plants under abiotic stress usually trigger hormonal signaling pathways. Abscisic acid (ABA) and ethylene, the most frequent stimulants of abiotic responses, also interact with other phytohormones [salicylic acid (SA), JA, and auxin], altering gene expression and adaptive stress responses [38]. Auxin is distributed in plant tissues helping plants to adapt to environmental conditions by controlling biological processes [39].

In addition, many developmental processes and physiological responses, including cell division, floral development, and hormonal signaling, are influenced by proteins after post-translational control, even protein turnover by ubiquitination [40]. Concerning the ubiquitination process, the substrate specificity derives from the E3 ubiquitin-ligase enzymes. The Cullin-RING Ligases (CRLs) are E3 Ligases binding to proteins to be ubiquitinated. The gene encoding the cullin-like protein1 (Manes.02G033400) presented six SNPs/INDELs of low impact (S2 Table). Furthermore, the F-box proteins confer the substrate specificity of the SCF complex, and the CUL1 association involves the SKP1 adapter protein. The SCF$^{TIR1/AFB}$ complex acts as an auxin receptor, and binding with auxin increases the affinity of the complex for substrates, such as Aux/IAA proteins, increasing the ubiquitination and protein turnover. As a result, auxin response factors (ARFs), normally repressed by dimerization with Aux/IAAs, are released, acting as transcriptional regulators of auxin-responsive genes [41]. Besides, the signalosome COP9 (Constitutive Photomorphogenesis 9), known as CSN, cleaves RUB/NEDD8 from Cullins [42]. Thus, CSN acts as a promoter of E3 Ligase activity by protecting the substrate adapters of CRLs from autocatalytic degradation [43]. From the predicted PPI network, the components RBX1 (from the SCF ubiquitination complex for AUX response), CAND1 (Cullin-associated and neddylation-dissociated protein 1), and Cullin 1 (Manes.02G033400) have been reported for interacting with COP9 and the CSN complex [44]. Also, the role of CSN in the JA biosynthesis pathway has already been reported [45]. Another group directly related to the COP9 complex involved FUS12 (CSN2 - subunit 2 of the signalosome COP9 complex), a recognized repressor of plant development [46], and ARF6 protein (auxin response factor 6; Manes. 07G102300), also interacting with SHY2 (Short hypocotyl 2, an Aux-IAA protein).

Since environmental abiotic stresses also induce alternative transcripts [47], a potential mRNA splicing factor (Manes.01G115300) was linked with proteins including MEE (nuclear component ribonucleoprotein CLO-5), MAC (pre-mRNA processing factor 19 homologous 1), SKIP (protein-SNW/SKI interaction), PRP (protein pre-mRNA splicing processing), CDC (cell division cycle protein), and ESP (pre-mRNA-splicing factor RNA helicase DEAH1). All these proteins characterized by GO terms highlighted alternative splicing (AS) of RNAs. Polymorphisms covering the gene encoding that putative mRNA splicing factor (Manes.01G115300) comprised three SNPs/INDELs; one of moderate effect (missense variant) would be heterozygous in bitter BRS Formosa and homozygous in BRS Dourada (S2 Table).

Additional interactions involved the protein PAP15 (purple acid phosphatase 15, Manes.15G054300), already reported to enhance plant tolerance to osmotic and salinity stresses by modulation of ascorbic acid (AsA) levels, an antioxidant compound effective against the overproduced ROS [48]. The related gene presented six SNPs/INDELs of low impact and two of moderate effects (S2 Table). One missense variant of moderate effect was heterozygous in the BRS Formosa and non-polymorphic in BRS Dourada (S2 Table). Again, the heterozygous condition was evidenced in the bitter (BRS Formosa) variety, to the detriment of the homozygous or non-polymorphic status in the sweet BRS Dourada.

**II. The PPI network based on BRS Formosa's hypermethylated genes.** The hypermethylated genes of the bitter BRS Formosa potentially silenced three PPI networks, one of

them linked several proteasome-related proteins. As proteins control cellular activities and physiological processes through the regulation of specific biochemical or metabolic pathways, in this regulation, protein degradation is as relevant as protein synthesis. Proteases and the ubiquitin-proteasome system (UPS) are responsible for the most protein degradation (80 - 90%), and the UPS system is an efficient and fast strategy to control cellular processes by selectively removing regulatory proteins [49]. Also, genes encoding proteases are induced in plants under abiotic stress, altering proteolytic activity levels of different specificities [50].

Other network members potentially silenced, but not expected to be, were transcription initiation factor TFIID subunits. These proteins are part of the RNA polymerase II pre-initiation complex. The promoter recognition by the basal transcription factor TFIID includes the TATA-binding protein (TBP) in a complex with TBP-associated factors (TAFs). The TAFs are transcriptional co-activators, assembling transcriptional complexes and recognizing promoters [51]. Some TFIID-related TFs were reported to be ethylene-responsive and relevant in drought-stress responses [52]. Since the related genes were hypermethylated, the bitter BRS Formosa is not adequately exploring this alternative.

The adequate importation of proteins to the nucleus is relevant to gene expression reprogramming, leading to plant drought tolerance. In this way, a network member was the KPNB1 protein (importin beta-1 subunit, ARM-repeat family, Manes.01G040400). The inactivation of the *AtKPNB1* gene from *A. thaliana* increased stomatal closure (in response to ABA influence), reduced water loss, and increased water deficit tolerance [53]; such results reinforced our finding of the hypermethylated gene.

Another potentially silenced PPI network stood out RNA regulation, epigenetic regulation, and gene silencing. Predicted interactions linked a member of the 3'-5'- exoribonuclease family (Manes.01G181600) and proteins with RNA degradation activity (RRP). A related gene (*AtRRP6L1*) promoting a loss-of-function mutation decreased DNA methylation, probably retaining RNAs and accumulating RNAs in association with chromatin [54]. Additional predicted proteins included RRP45b (subunit of the exosome) and CCR4-1, a relevant protein of the 3'-5' repression complex for RNA degradation [55].

## The sweet BRS Dourada's interactome

Concerning the sweet BRS Dourada, plant strategies derived from the hypomethylated genes included ribosomal biogenesis, regulation of ABC family transporters, root development/plant growth, and the DNA repair system. From the hypermethylated genes, the plant strategies potentially silenced included cell proliferation, RNA regulation, and protein translation/synthesis, as detailed below.

**I. The PPI networks based on BRS Dourada's hypomethylated genes.** Concerning the proteins encoded by hypomethylated genes of the sweet BRS Dourada, small PPI networks stood out. A network linked a transporter of the ABC family (ABCF1, Manes.02G005700), a Lysine -tRNA ligase (Manes.10G055800), and two 40S ribosomal proteins (S8-2, Manes.03G134200; S9-2, Manes.05G099700) with proteins/ribosomal subunits. In plants under (a)biotic stress, genes encoding ribosomal proteins (RPs) are differentially regulated, affecting ribosomal biogenesis and plant growth [56]. Mutations in *RP*-related genes have regulatory roles in plant development processes, showing the mutants some abnormalities and reduced cell growth/proliferation [57]. The analysis of the related genes, besides showing several polymorphisms, also detected the heterozygous condition of the sweet BRS Dourada variety in detriment of the homozygous or non-polymorphic status of BRS Formosa, as observed for the genes encoding Lysine -tRNA ligase (Manes.10G055800) and 40S ribosomal protein S8-2 (Manes.03G134200), both showing one SNP/INDEL of moderate impact (a missense variant) (S2 Table).

Regarding plant water use efficiency, some RPL proteins (60S ribosomal proteins) were already reported as relevant [58], and such proteins (AT2G17360, AT5G58420, AT5G67510, AT2G20450, and AT3G49910) appeared In the PPI network. Mutant plants for the *RPL11* gene regulated by salt stress showed defective growth [59], while mutant plants for the *AtRPL24* gene (*A. thaliana*) had their development affected, as well as their restart of mRNA translation of *bZIP11* and *ARF*, both TFs relevant in abiotic stress responses [60]. In addition to the RPs, some interactions linked the PWP protein [periodic tryptophan (W) protein], the EMB complex [nucleolar associated protein complex (NOC4), and the 40S ribosomal protein S6-2 (RPSb)]. PWP proteins participate in ribosome biogenesis (18S ribosomal pre-RNA processing), while NOC4 proteins contribute to nucleolar processing of the 40S/90S rRNAs, maturation of 5,8S rRNA, and ribosome assembly [61].

Another component already reported in plants responding to (a)biotic stress [62] and predicted network member was the ABCF1 transporter (Manes.02G005700), which related gene (*LrABCF1*, *Lilium regale*) reinforced its involvement in addition to transmembrane transport inducing transcription in plants under abiotic stimuli (cold, salinity, and wounds) after SA and ethylene (ET) treatment.

Additional interactions highlighted the DNA mismatch repair protein MSH2 (Manes.01G020600) with other proteins acting in DNA damage repair (MSH, MLH, and PMS1). Components of the MMR (mismatch repair) system form MutS alpha heterodimers (MSH2-MSH6 heterodimer) that bind to the DNA mismatches, initiating the repair. These proteins correct base-base mismatches and insertion-deletion loops from eventual DNA damages [63]. The DNA repair keeps the plant metabolism working properly [64].

Also, interactions linked proteins from the SCAR and the Arp2/3 complexes. The ARP proteins play a key role in actin polymerization, mediating the formation of branched networks of actin. Furthermore, the Arp2/3 complex activates the SCAR complex, staying directly involved in plant growth [65].

**II. The PPI networks based on BRS Dourada's hypermethylated genes.** Proteins probably silenced by the hypermethylated genes from the sweet BRS Dourada presumed five networks. One network linked the 2-oxoglutarate dehydrogenase (component E1; Manes.10G034400) with the domain protein RING associated with BRCA1 (Manes.12G035400), a member of the mini-chromosome maintenance (MCM) family protein (Manes.17G068400), and an unknown protein (Manes.05G206400). Such proteins interact in origin of replication complexes (ORC) and involve cell division cycle (CDC) proteins. The MCM proteins participate in the initiation of DNA replication (transition from G1 to S phase). Such a phase requires the formation/activation of the pre-replicative complex, which begins with the binding of CDC6 to ORCs during the G1 phase, allowing the recruitment of the MCM protein complex. In turn, the S phase is triggered by the activation of this complex by cyclin-dependent kinases, which leads to a change from the pre-replication to a post-replication complex [66]. The hypermethylation of the related genes will reduce the cell proliferation after the stress.

Other interactions linked the DEAD-box ATP-dependent RNA helicase 47 (mitochondrial, Manes.02G025500) with translation initiation factor 3 (Manes.02G056600) and members of the eIF family. The eIFs (families eIF1 to eIF6) act in translation initiation and protein synthesis [67]. The eIF1, eIF1A, and eIF3 form the 43S pre-initiation complex (PIC) through the binding of ribosomal subunits and in turn, eIF4 promotes the correct combination of PIC with the CAP-5′ end [68]. Some eIF genes have already been reported in plants responding to abiotic stresses. Genes such as *MieIF1A-a*, *MieIF5*, and *MieIF3sB* were strongly expressed in plants under salinity, drought (PEG), and cold stress, respectively [68]. Since some of the

mentioned proteins involved hypermethylated genes, the sweet BRS Dourada probable does not explore efficiently that alternative.

Additional interaction linked a member of the CPSF complex (cleavage specification factor and polyadenylation, Manes.10G123600) with the PAP (nuclear poly(A)-polymerase) protein. The CPSF proteins play a key role in 3'-end pre-mRNA formation by interacting with poly (A)-polymerase and protein factors to cleave the transcript and add the poly(A) tail. Such proteins could be involved in the post-transcriptional control of plants responding to stress as highlighted during oxidative stress [69]. Another interaction linked protein of the RNA exosome complex (RRP45B, Manes.01G181600) to a member of the ARM repeat superfamily protein (Manes.15G056200). The RNA exosome complex is a multiprotein complex that targets RNAs acting from maturation to quality control and final turnover. Additional interaction involved a PWI domain splicing factor (Manes.01G169300) with the 4-a subunit of the THO complex (Manes.04G032200). The PWI domain in splicing factors is the RNA/DNA binding domain and it has multiple functions in pre-mRNAs processing [70]. In turn, the THO complex, subunit 4, participates in the biogenesis of the ribonucleoprotein particle (mRNP) that aids in the export of mRNAs out of the nucleus [71]. Therefore, the RNA regulation could be affected in the BRS Dourada stress response. Thus, the potential hypermethylation of those related genes could compromised the RNA regulation and also the plant stress-response.

## Differentially methylated genes in transcriptomic studies in cassava

The search for potential DMGs identified in this study, classified as hyper- or hypo-methylated, in cassava transcriptomics studies showed that of the 190 candidates, 32 of them had differential gene expression reported in at least three plant (cassava) assays after exposure to abiotic stress; two of them exposure to PEG 6000 (20%) and cold (4 oC), reported by Li et al. (2017) [72] and another, also of exposure to PEG 6000 (20%), authored by Ding et al. (2019) [73, 74]. In all trials, the hyper-methylated candidates were classified exclusively as down-regulated, while the hypo-methylated candidates were all declared up-regulated, and only for one of them the up-regulation expression was found in one of the trials, but in another trial the down-regulation, expression was reported. The candidates, methylation level and differential gene expression in each assay are available in S8 Table.

## Final considerations

In summary, our analysis of methylated tags in the *M. esculenta* genome provides valuable insights revealing several noteworthy findings. We found consistent data representation across two different varieties supported by Pearson's correlation coefficients for biological replicates. Methylated sites were primarily concentrated within the initial 20% of gene lengths. Notably, DMGs showed a higher prevalence of methylated CCGG sites compared to CCNGG sites in both varieties. Importantly, the BRS Formosa and BRS Dourada varieties exhibited distinct methylation profiles with limited overlap in DMGs. The gene ontology characterization and PPI analysis supported these observations with BRS Formosa's methylome indicating a focus on drought-stress response and BRS Dourada's methylome showing heightened sensitivity to drought. BRS Formosa displayed a more immediate and effective stress response involving a repression of primary metabolism and plant development along with induction of stress-signaling pathways. In contrast, BRS Dourada emphasized the regulation of plant development and induced the DNA repair system.

We also examined SNPs and insertions/deletions (INDELs) in genes within the interactomes revealing genetic variations and insights into the condition of heterozygosis/homozygosis. These variations hold promise for further studies on drought-stress tolerance. Ongoing

investigations into the genomes and transcriptomes of these cassava varieties will provide a more comprehensive understanding of genetic divergence and its relationship with the observed methylome profiles. This research significantly advances our knowledge of methylome variation in these contrasting BRS varieties following water deficit stress offering valuable insights for cassava breeding programs focused on developing drought-tolerant cassava elite materials.

## Supporting information

**S1 Fig. The two-way effect (soil water deficit and genotype) on gas exchange in plants subjected to a 15-day period of soil water drying.**
(TIF)

**S2 Fig. The one-way effect of genotypes on plant gas exchange (A, gs, E) during the 15-day period of soil water drying.**
(TIF)

**S3 Fig. The one-way effect of soil water deficit on gas exchange in cassava plants.**
(TIF)

**S4 Fig. Distribution of methylation sites across genes, based on RPM data.** The X axis represents the relative position in the gene (i.e. 20 means 20% of the initial positions of the gene). A: analysis for CCGG sites; B: analysis for CCNGG sites. Legend: A01: FST-R1; A02: FST-R2; A03: FST-R3; A04: FNC-R1; A05: FNC-R2; A06: FNC-R3; A07: DST-R1; A08: DST-R2; A09: DST-R3; A10: DNC-R1; A11: DNC-R2; A12: DNC-R3; R1, R2 and R3: biological triplicates; FST - BRS Formosa Stressed Treatment; FNC – BRS Formosa Negative Control; DST – BRS Dourada Stressed Treatment; DNC – BRS Dourada Negative Control.
(TIF)

**S5 Fig. Distribution of methylation sites along transcription initiation regions (TSS) indicated by the zero point on the x-axis. RPM count data.** The x-axis represents the 2000 bp window before and after the start of transcription. Graph A represents the analysis for CCGG site; B represents analysis for CCNGG site. A01: FST-R1; A02: FST-R2; A03: FST-R3; A04: FNC-R1; A05: FNC-R2; A06: FNC-R3; A07: DST-R1; A08: DST-R2; A09: DST-R3; A10: DNC-R1; A11: DNC-R2; A12: DNC-R3; R1, R2 and R3: biological triplicates; FST - BRS Formosa Stressed Treatment; FNC – BRS Formosa Negative Control; DST – BRS Dourada Stressed Treatment; DNC – BRS Dourada Negative Control.
(TIF)

**S6 Fig. Distribution of methylation sites along transcription termination regions (TTS) indicated by the zero point on the x-axis. RPM count data.** The x-axis represents the 2000 bp window before and after the start of transcription. Graph A represents the analysis for CCGG site; B represents analysis for CCNGG site A01: FST-R1; A02: FST-R2; A03: FST-R3; A04: FNC-R1; A05: FNC-R2; A06: FNC-R3; A07: DST-R1; A08: DST-R2; A09: DST-R3; A10: DNC-R1; A11: DNC-R2; A12: DNC-R3; R1, R2 and R3: biological triplicates; FST - BRS Formosa Stressed Treatment; FNC – BRS Formosa Negative Control; DST – BRS Dourada Stressed Treatment; DNC – BRS Dourada Negative Control.
(TIF)

**S7 Fig. Pearson correlation plot for CCGG site of each replica.** The upper left triangle is the scatterplot of the methylation level between two samples that are diagonally across the image, the lower right triangular region is the corresponding Pearson correlation coefficient. A01:

FST-R1; A02: FST-R2; A03: FST-R3; A04: FNC-R1; A05: FNC-R2; A06: FNC-R3; A07: DST-R1; A08: DST-R2; A09: DST-R3; A10: DNC-R1; A11: DNC-R2; A12: DNC-R3; R1, R2 and R3: biological triplicates; FST - BRS Formosa Stressed Treatment; FNC – BRS Formosa Negative Control; DST – BRS Dourada Stressed Treatment; DNC – BRS Dourada Negative Control.
(TIF)

**S8 Fig. Pearson correlation plot for CCNGG site of each replica.** The upper left triangle is the scatterplot of the methylation level between two samples that are diagonally across the image, the lower right triangular region is the corresponding Pearson correlation coefficient. A01: FST-R1; A02: FST-R2; A03: FST-R3; A04: FNC-R1; A05: FNC-R2; A06: FNC-R3; A07: DST-R1; A08: DST-R2; A09: DST-R3; A10: DNC-R1; A11: DNC-R2; A12: DNC-R3; R1, R2 and R3: biological triplicates; FST - BRS Formosa Stressed Treatment; FNC – BRS Formosa Negative Control; DST – BRS Dourada Stressed Treatment; DNC – BRS Dourada Negative Control.
(TIF)

**S9 Fig.** Heatmap graph of methylated genes (p-value $\leq$ 0.05 and log2FC > 1) in BRS Formosa variety, presented by the biological replicas (R1, R2 and R3) in drought condition or control without stress, based on log2 of RPM; A01 (FST-R1); A02 (FST-R2); A03 (FST-R3); A04 (FNC-R1); A05 (FNC-R2); A06 (FNC-R3); FST – BRS Formosa Stressed Treatment, FNC – BRS Formosa Negative Control; A) hypermethylated genes in CCGG sites; B) hypomethylated genes in CCGG sites; C) hypermethylated genes in CCNGG sites; D) hypomethylated genes in CCNGG sites.
(TIF)

**S10 Fig.** Heatmap graph of methylated genes (p-value $\leq$ 0.05 and log2FC > 1) in BRS Dourada variety, presented by the biological replicas (R1, R2 and R3) in drought condition or control without stress, based on log2 of RPM; A07 (DST-R1); A08 (DST-R2); A09 (DST-R3); A10 (DNC-R1); A11 (DNC-R2); A12 (DNC-R3); DST – BRS Dourada Stressed Treatment, DNC – BRS Dourada Negative Control; A) hypermethylated genes in CCGG sites; B) hypomethylated genes in CCGG sites; C) hypermethylated genes in CCNGG sites; D) hypomethylated genes in CCNGG sites.
(TIF)

**S1 Table. Differentially methylated genes.** Columns indicate, respectively: studied contrast BRS Formosa Stressed Tratament vs BRS Formosa Negative Control (FST vs FNC) or BRS Dourada Stressed Tratament vs BRS Dourada Negative Control (DST vs DNC); methylation site (CCGG or CCNGG), differentially detected methylation (Hypo- or Hypermethylated); gene identifier, and annotation based on the v.6.1 genome from the Phytozome database.
(XLSX)

**S2 Table. Differentially methylated genes in BRS Dourada and BRS Formosa varieties, their identifiers and annotations, methylation status, polymorphims events according their impacts (low, moderate, high), and descriptions of a selected exemple: Chromossome (based on the Manihot esculenta genome v6.1 of Phytozome v12.1); Position (of the polymorphism); Ref.** (nucleotide in the ref. genome), Polymorphism (nucleotide/sequence detected in the BRS variety); Impact (Low, Moderate, High); Consequence (polymorphism type); Heterozygous/Homozygous status; sequences in FASTA format.
(XLSX)

**S3 Table. GO terms enriched (p-value < 0.01) from hypo- and hypermethylated genes for BRS Formosa Stressed Tratament vs BRS Formosa Negative Control (FST vs FNC) and BRS Dourada Stressed Tratament vs BRS Dourada Negative Control (DST vs DNC).** The columns indicate, respectively: Contrast (FST vs FNC or DST vs DNC); gene methylation type (Methylation); id of term GO (GO.ID); description of the term GO (Term); analysis confidence parameters (expected, p-value, q-value); Aspect (P: biological process; C: cellular component; F: molecular function); and genes inserted in the analysis that have the term GO (Genes).
(XLSX)

**S4 Table. GO and KEGG terms for proteins of the interatoma formed by differentially hypomethylated sites for the bitter variety BRS Formosa based on *A. thaliana* proteins orthologous to *M. esculenta* proteins.** The columns indicate respectively: Id of the GO term our KEGG term (Term ID); term description; Aspect (P: biological process; C: cellular component; F: molecular function; K: KEGG terms); term count in the interactome (Gene count); information background for this term in the database (Background gene count); reliability level (Strength); False discovery rate; identifier of proteins containing the GO or KEGG term (Matching proteins 1); name of the proteins that contain the GO or KEGG term (Matching proteins 2).
(XLSX)

**S5 Table. GO and KEGG terms for proteins of the interatoma formed by differentially hypermethylated sites for the bitter variety BRS Formosa based on *A. thaliana* proteins orthologous to *M. esculenta* proteins.** The columns indicate respectively: Id of the GO term our KEGG term (Term ID); term description; Aspect (P: biological process; C: cellular component; F: molecular function; K: KEGG terms); term count in the interactome (Gene count); information background for this term in the database (Background gene count); reliability level (Strength); False discovery rate; identifier of proteins containing the GO or KEGG term (Matching proteins 1); name of the proteins that contain the GO or KEGG term (Matching proteins 2).
(XLSX)

**S6 Table. GO and KEGG terms for proteins of the interatoma formed by differentially hypomethylated sites for the sweet variety BRS Dourada based on *A. thaliana* proteins orthologous to *M. esculenta* proteins.** The columns indicate respectively: Id of the GO term our KEGG term (Term ID); term description; Aspect (P: biological process; C: cellular component; F: molecular function; K: KEGG terms); term count in the interactome (Gene count); information background for this term in the database (Background gene count); reliability level (Strength); False discovery rate; identifier of proteins containing the GO or KEGG term (Matching proteins 1); name of the proteins that contain the GO or KEGG term (Matching proteins 2).
(XLSX)

**S7 Table. GO and KEGG terms for proteins of the interatoma formed by differentially hypermethylated sites for the sweet variety BRS Dourada based on *A. thaliana* proteins orthologous to *M. esculenta* proteins.** The columns indicate respectively: Id of the GO term our KEGG term (Term ID); term description; Aspect (P: biological process; C: cellular component; F: molecular function; K: KEGG terms); term count in the interactome (Gene count); information background for this term in the database (Background gene count); reliability level (Strength); False discovery rate; identifier of proteins containing the GO or KEGG term (Matching proteins 1); name of the proteins that contain the GO or KEGG term (Matching

proteins 2).
(XLSX)

**S8 Table. Differentially Methylated Gene (DMG) and gene expression data.** Columns indicate, respectively: Cassava variety (bitter BRS Formosa or sweet BRS Dourada); methylation site (CCGG or CCNGG), methylation status (hypo- or hypermethylation); Gene ID (Manihot esculenta v6.1, Phytozome) and annotation.
(XLSX)

## Author Contributions

**Conceptualization:** Jorge Luís Bandeira da Silva Filho, Claudia Fortes Ferreira, Eder Jorge de Oliveira, Ederson Akio Kido.

**Data curation:** Jorge Luís Bandeira da Silva Filho, Wilson José da Silva Júnior.

**Formal analysis:** Wilson José da Silva Júnior.

**Funding acquisition:** Eder Jorge de Oliveira.

**Investigation:** Jorge Luís Bandeira da Silva Filho, Rosa Karla Nogueira Pestana.

**Methodology:** Jorge Luís Bandeira da Silva Filho, Rosa Karla Nogueira Pestana, Maurício Antônio Coelho Filho, Claudia Fortes Ferreira, Eder Jorge de Oliveira, Ederson Akio Kido.

**Project administration:** Claudia Fortes Ferreira, Eder Jorge de Oliveira, Ederson Akio Kido.

**Resources:** Eder Jorge de Oliveira.

**Software:** Wilson José da Silva Júnior.

**Supervision:** Claudia Fortes Ferreira, Eder Jorge de Oliveira.

**Visualization:** Jorge Luís Bandeira da Silva Filho, Maurício Antônio Coelho Filho.

**Writing – original draft:** Jorge Luís Bandeira da Silva Filho, Ederson Akio Kido.

**Writing – review & editing:** Rosa Karla Nogueira Pestana, Maurício Antônio Coelho Filho, Claudia Fortes Ferreira, Eder Jorge de Oliveira.

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
