## [Decision Letter · Decision Letter 0]

10 Oct 2023

PONE-D-23-29799Exploiting DNA methylation in cassava under water deficit for crop improvementPLOS ONE

Dear Dr. de Oliveira,

Thank you for submitting your manuscript to PLOS ONE. After careful consideration, we feel that it has merit but does not fully meet PLOS ONE’s publication criteria as it currently stands. Therefore, we invite you to submit a revised version of the manuscript that addresses the points raised during the review process.

We look forward to receiving your revised manuscript.

Kind regards,

Evangelia V. Avramidou, PhD

Academic Editor

PLOS ONE

4. We note that Figure 2 in your submission contain copyrighted images. All PLOS content is published under the Creative Commons Attribution License (CC BY 4.0), which means that the manuscript, images, and Supporting Information files will be freely available online, and any third party is permitted to access, download, copy, distribute, and use these materials in any way, even commercially, with proper attribution. For more information, see our copyright guidelines: http://journals.plos.org/plosone/s/licenses-and-copyright.

1. You may seek permission from the original copyright holder of Figure 2 to publish the content specifically under the CC BY 4.0 license.

Additional Editor Comments:

Dear authors,

please reply to reviewer's comments in order to answer their comments.

With kind regards

Reviewers' comments:

Reviewer's Responses to Questions

**Comments to the Author**

1. Is the manuscript technically sound, and do the data support the conclusions?

Reviewer #1: Yes

Reviewer #2: Yes

2. Has the statistical analysis been performed appropriately and rigorously? 

Reviewer #1: Yes

Reviewer #2: Yes

3. Have the authors made all data underlying the findings in their manuscript fully available?

Reviewer #1: Yes

Reviewer #2: Yes

4. Is the manuscript presented in an intelligible fashion and written in standard English?

Reviewer #1: Yes

Reviewer #2: No

5. Review Comments to the Author

Reviewer #1: The paper entitled "Exploiting DNA methylation in cassava under water deficit for crop improvement" aim to provide information regarding genome regions of cassave that are hypo or hypermethylated under water deficit and the implication of changes in DNA methylation status in drought tolerace.

The paper is interesting however some issues need to be clarified, especially in the materials and methods section.

1. Apart form the photos of the plants after the completion of drought stress (with claer differences in morphology) and the Transpirable Soil Water (FTSW) calculation, no other morphological or physiological measurements are described.

Can you provide some more data regarding the differences in morphology and physiology og the 2 genotypes after drought stress?

Materials and methods section needs to be divided into sections, first describing more clearly the experimental design and the duration of the drought stress and then focusing on the molecular studies.

2. Conclusion section needs to be rewritten in some parts and need to give more realistic outcomes.

3. please see the attachment and provide answers accordingly.

Reviewer #2: The study is particularly important and original. The methodology followed is detailed, with many analyzes and interpretations of the results. I want just a few clarifications on the methodology. What were the prevailing conditions in the greenhouse, in which the experiment was carried out? Why were only 5 plants per variety selected? Why are only the parents of the variety BRS Formosa listed and not both varieties.

6. PLOS authors have the option to publish the peer review history of their article (what does this mean?). If published, this will include your full peer review and any attached files.

Reviewer #1: No

Reviewer #2: No

---

## [Author Response · Author response to Decision Letter 0]

3 Nov 2023

Response reviewers

Response: All data sets are available at the link <https://doi.org/10.6084/m9.figshare.21330708.v2>, which is provided in the 'Data Availability Statement' section.

Response: We have revised our text. All the information is available in the manuscript and the link provided in the 'Data Availability Statement' section 

4. We note that Figure 2 in your submission contain copyrighted images. All PLOS content is published under the Creative Commons Attribution License (CC BY 4.0), which means that the manuscript, images, and Supporting Information files will be freely available online, and any third party is permitted to access, download, copy, distribute, and use these materials in any way, even commercially, with proper attribution. For more information, see our copyright guidelines: http://journals.plos.org/plosone/s/licenses-and-copyright.

Response: Figure 2 does not contain copyrighted images. The authors themselves have provided this figure, which will now be included in the respective document.

Reviewer #1: 

The paper is interesting however some issues need to be clarified, especially in the materials and methods section.

1. Apart form the photos of the plants after the completion of drought stress (with claer differences in morphology) and the Transpirable Soil Water (FTSW) calculation, no other morphological or physiological measurements are described. Can you provide some more data regarding the differences in morphology and physiology og the 2 genotypes after drought stress?

Response: Additional information was added at the beginning of the 'Results' section. Please let us know if you have any remaining questions.

Materials and methods section needs to be divided into sections, first describing more clearly the experimental design and the duration of the drought stress and then focusing on the molecular studies.

Response: Response: We appreciate your feedback. Acknowledging your comment, we recognize that restructuring our methodology into sections enhanced readability and understanding. We have corrected this aspect accordingly

2. Conclusion section needs to be rewritten in some parts and need to give more realistic outcomes.

Response: Thank you for the suggestion regarding the conclusion of our manuscript. We've restructured it to be more concise and direct, while also incorporating additional relevant information about our obtained results.

3. please see the attachment and provide answers accordingly.

Response: Ok, thank you. 

Reviewer #2: 

The study is particularly important and original. The methodology followed is detailed, with many analyzes and interpretations of the results. I want just a few clarifications on the methodology. What were the prevailing conditions in the greenhouse, in which the experiment was carried out? 

Response: Additional information has been included within the 'Materials and Methods' section to address these queries.

Why were only 5 plants per variety selected? 

Response: There were 5 plants per plot. Please review the modifications within the 'Materials and Methods' section.

Why are only the parents of the variety BRS Formosa listed and not both varieties.

Response: Please check the following sentence in the text.

“BRS Dourada is a local variety with an unknown pedigree, specifically developed for fresh consumption.”

---

## [Editor Report · Decision Letter 1]

8 Dec 2023

Exploiting DNA methylation in cassava under water deficit for crop improvement

PONE-D-23-29799R1

Dear Dr. de Oliveira,

We’re pleased to inform you that your manuscript has been judged scientifically suitable for publication and will be formally accepted for publication once it meets all outstanding technical requirements.

Kind regards,

Evangelia V. Avramidou, PhD

Academic Editor

PLOS ONE

Additional Editor Comments (optional):

Dear authors,

all comments have been adressed, and the manuscript was significantly improved and it is ready for publication.
---

## [Editor Report · Acceptance letter]

14 Dec 2023

PONE-D-23-29799R1 

PLOS ONE

Dear Dr. de Oliveira, 

I'm pleased to inform you that your manuscript has been deemed suitable for publication in PLOS ONE. Congratulations! Your manuscript is now being handed over to our production team.

Kind regards, 

on behalf of

Dr. Evangelia V. Avramidou 

Academic Editor

PLOS ONE